# Navigating the Landscape: A Comprehensive Review of Current Virus Databases

**DOI:** 10.3390/v15091834

**Published:** 2023-08-29

**Authors:** Muriel Ritsch, Noriko A. Cassman, Shahram Saghaei, Manja Marz

**Affiliations:** 1RNA Bioinformatics and High-Throughput Analysis, Friedrich Schiller University Jena, 07743 Jena, Germany; evbc@uni-jena.de; 2European Virus Bioinformatics Center, 07743 Jena, Germany; 3German Centre for Integrative Biodiversity Research (iDiv) Halle-Jena-Leipzig, 04103 Leipzig, Germany; 4FLI Leibniz Institute for Age Research, 07745 Jena, Germany

**Keywords:** viruses, databases, genomes, sequences, metadata, FAIR evaluation

## Abstract

Viruses are abundant and diverse entities that have important roles in public health, ecology, and agriculture. The identification and surveillance of viruses rely on an understanding of their genome organization, sequences, and replication strategy. Despite technological advancements in sequencing methods, our current understanding of virus diversity remains incomplete, highlighting the need to explore undiscovered viruses. Virus databases play a crucial role in providing access to sequences, annotations and other metadata, and analysis tools for studying viruses. However, there has not been a comprehensive review of virus databases in the last five years. This study aimed to fill this gap by identifying 24 active virus databases and included an extensive evaluation of their content, functionality and compliance with the FAIR principles. In this study, we thoroughly assessed the search capabilities of five database catalogs, which serve as comprehensive repositories housing a diverse array of databases and offering essential metadata. Moreover, we conducted a comprehensive review of different types of errors, encompassing taxonomy, names, missing information, sequences, sequence orientation, and chimeric sequences, with the intention of empowering users to effectively tackle these challenges. We expect this review to aid users in selecting suitable virus databases and other resources, and to help databases in error management and improve their adherence to the FAIR principles. The databases listed here represent the current knowledge of viruses and will help aid users find databases of interest based on content, functionality, and scope. The use of virus databases is integral to gaining new insights into the biology, evolution, and transmission of viruses, and developing new strategies to manage virus outbreaks and preserve global health.

## 1. Introduction

Viruses, the most abundant and diverse biological entities in the biosphere, play important roles in public health, ecology, and agriculture [1,2]. The identification, characterization, and surveillance of viruses rely heavily on knowledge of their various characteristics, including genome organization, genomic sequences, and replication strategy [3,4,5]. Until about the 1990s, virus genomes were mainly derived from isolates, came from medically and agriculturally important hosts, and were annotated with morphological and medically related metadata [6]. With the advancements in and lower costs of next-generation sequencing technologies, -omics projects have gained popularity across various fields, leading to the continuous deposition of metagenome-derived viral sequences into sequencing repositories along with computationally derived metadata. The analysis of uncultivated virus genomes derived from metagenomic sequences has accelerated the rate of discovery of new viruses [7]. However, despite these technological strides, the current understanding of virus diversity remains incomplete [8,9]; for example, regarding viruses with zoonotic potential, one estimate suggests that only 1% of these virus species have been discovered to date [10]. This discrepancy highlights the need to further explore and uncover the vast array of undiscovered viruses through their genomic sequences.

Web-facing databases provide the structured and indexed storage of information, which can be easily accessed through an internet browser. In the context of virus research, virus databases represent central hubs of information connecting virus genomic sequences and associated metadata, such as viral and host taxonomy, host range, transmission modes, genomic structures, and gene annotations. Researchers use these databases to gain insights into viral genetic diversity and evolutionary relationships, and to study new viruses [11,12]. Virus databases play a pivotal role in virus discovery, surveillance, and monitoring, allowing for the comparison of newly identified viruses with known ones [13,14,15]. Moreover, they contribute to epidemiological studies by integrating diverse information, such as clinical characteristics, virus and host genetic variation, and samplings and geographical location. Overall, virus databases are vital resources for comprehensive analyses, comparative studies, and generating innovative strategies in virus detection, prevention, and control.

The presence of multiple virus databases can be attributed to variations in specialization, data types, and aims. Some databases are created for a specific research purpose or to support a specific virus research area, e.g., virus ecology or epidemiology; alternately, some databases focus on specific viruses or cover a wide range of viruses. Further, databases have different goals, ranging from information dissemination to providing web-based tools for data analysis. Database longevity refers to the ability of a database to remain functional and accessible over a long period of time, even as technology and software evolve [16]. Ensuring longevity for virus sequence databases includes regular maintenance and updates, the usage of standardized data formats, the creation of backups and archives, the implementation of open data policies, collaboration, funding, and trust, and usage by the community [17,18]. The current varied landscape of virus databases largely reflects the informational needs and funding of different types of virus research.

Many virus databases feature the extra curation of a subset of virus sequences from long-term sequence repositories (e.g., NCBI Genbank) as well as additional computed datasets based on these sequences. As such, virus databases contain vital metadata, such as the taxonomic details of hosts and viruses, location data, publications, years, processing information, morphology, and more. These metadata play a crucial role in studying virus outbreaks, tracking infection pathways, and analyzing viral distribution patterns. Another important type of data found in virus databases is gene annotations, providing valuable information about genetic elements, protein sequences, gene functions, and structural features. These annotation data allow researchers to analyze the functions of viral genes and proteins, gaining insights into their impact on virus biology. These datasets facilitate the comparative analysis of viruses and virus communities. On the other hand, incomplete or inaccurate metadata in sequence entries can adversely impact downstream analysis [19,20]. Virus databases also offer analysis tools and software, such as sequence comparisons, phylogenetic analyses, protein structure prediction, and the identification of evolutionary selection pressures. A comprehensive comparison of the content based on the number of sequences and species in each database has not yet been performed.

A challenge encountered by all databases—not limited to virus-specific ones—is the handling of errors. Errors are almost unavoidable and can significantly impact subsequent analyses. Many questions arise when considering error management, such as whether users should be allowed to upload their own data, which can lead to a more complete dataset but also carries the potential for more errors. Another important aspect is how quickly and effectively errors can be addressed and supported. One solution to address this issue is to provide a curated subset of data, allowing users to decide which dataset to utilize. However, this presents difficulties, as users may not be aware of the cascading nature of existing errors. As far as we know, there is currently no comprehensive review systematically describing different types of errors to provide users with an overview of potential pitfalls so that users can better address them.

To the best of our knowledge, the last two comprehensive reviews of virus databases were conducted by Sharma et al. in 2015 and Mcleod et al. in 2017 [21,22]. Sharma et al. reviewed 60 tools and 50 databases, but 39 of the databases are no longer accessible or have not been updated recently; see Appendix A. In the Mcleod et al. review, they listed 149 entries, including 48 sequence databases. However, only 11 sequence databases met our criteria of being up to date for our study; see Appendix A. Comparing the two reviews, we found eight databases that were considered up to date in both. Surprisingly, despite the rapid advancements and numerous changes in the field, there has been a notable absence of comprehensive reviews on virus databases in the past 5 years.

In addition to database review articles, several catalogs of databases have been developed that aggregate databases in a clear and well-structured manner. Five such catalogs of databases are (1) re3data.org, (2) FAIRsharing [23], (3) The Database Commons [24], (4) ELIXIR bio.tools [25], and (5) NAR database list [26]. These catalogs may serve as valuable resources for users who are uncertain about which database to utilize, by providing them with a convenient starting point. In these database lists, users can find database descriptions and searchable metadata, as well as static properties, such as the year of establishment, links, and country information. There is currently a lack of comparisons of catalogs of databases evaluating their use for searching virus databases.

When using a virus database, users have diverse requirements beyond accessing relevant content. The usability of the database is crucial, with easy navigation and efficient information retrieval being key factors. Features like keyword search and phrase suggestions enable efficient entry retrieval. Presenting results in meaningful formats, such as tables, enhances usability, particularly when multiple matching entries are involved. Accessibility is important, providing options like one-click downloads and availability through various channels (website, FTP, or API), considering the need for user login. Establishing links to other databases enables comprehensive and integrated research, while integration with workbenches and the availability of associated tools empower users to work with their data and utilize additional resources for analysis. The easy sharing of URLs promotes collaboration and efficient communication. Trust and transparency are fostered by the availability of source code, enabling users to comprehend underlying processes. To the best of our knowledge, there are currently no existing reviews that attempt to describe the functionality of databases. This lack of information hinders users in making informed decisions about which database is the most suitable for their needs.

The FAIR data principles encompass the findability, accessibility, interoperability, and reusability of research data objects [27]. The principles emphasize making these objects FAIR (whether database entries or databases) by following the criteria within each principle described at the FAIR principles website. The FAIR principles can be applied to evaluate three key entities: data (or any digital object), metadata (information about the digital object), and infrastructure. Adhering to the FAIR principles is meant to enhance machine-driven discovery and the utilization of data in parallel to facilitating human collaboration and data reuse in scientific research. Evaluating virus databases based on FAIR principles would reveal their potential to advance scientific knowledge and enable data-driven discoveries. As far as our knowledge extends, this is the initial evaluation of virus databases in terms of FAIRness.

In this study, we provide a comprehensive overview of the most up-to-date virus databases, covering five key aspects: (1) content, (2) functionality, (3) FAIRness, (4) comparison of the catalogs of databases, and (5) an analysis of potential errors within the databases. In terms of content, we compare the key features already present in other reviews, such as name, link, last update, short description, and citations. Additionally, we delve deeper into the content of the databases by evaluating the number of sequences and species available, suggesting use cases for each database. Regarding functionality, we compare features such as general usage, presence of a workbench, tool availability, database complexity, download options and ease of use, and the need for user login. Additionally, we analyze whether keyword search functionality is available, whether there are phrase suggestions, the presentation format of the output, the presence of links to other databases, the ability to share URLs, and the availability of source code. Furthermore, we evaluate the FAIRness (findable, accessible, interoperable, and reusable) of the databases, as it is of paramount importance for their usability and long-term value. In our review, we include an up-to-date assessment of the webpages hosting virus databases, specifically focusing on (1) re3data.org, (2) FAIRsharing [23], (3) The Database Commons [24], (4) ELIXIR bio.tools [25], and (5) NAR database list [26]. Additionally, we provide an overview of various types of errors that can occur in databases, providing users with suggestions to better address and mitigate these issues during deposition of user data to the sequencing repositories.

## 2. Exploring the Distinctions amongst Diverse Databases

Here, we present a comprehensive overview of 24 virus databases that are currently active and contain virus-centric data (e.g., virus sequences); see Table 1 and Appendix A.

We compiled the list of databases for comparisons by including all up-to-date databases mentioned in the previous review papers [21,22], along with active databases obtained from the database catalogs (re3data.org, FAIRsharing, The Database Commons, ELIXIR bio.tools, and NAR database list).

We only examined up-to-date databases, meaning databases with a website or data update from 2022 onward. Due to the COVID-19 pandemic, numerous coronavirus-related databases have been developed and established. In total, we identified 49 such coronavirus databases, of which we selected and included 3 important ones in Table 1 based on our assessment. All other coronavirus databases identified here are listed in Appendix A. For further reference, please see a relatively recent review of COVID-19 resources [28]. Additionally, we excluded several databases based on various criteria. Some were considered too specific, focusing on a single virus in a particular region or providing exclusively epidemiological data. Others, we classified as tools, networks, or tables rather than databases.

The following non-COVID databases were found in our search of databases and were not included in our Table 1 based on the criteria for rejection listed above: (1) Disease Monitoring Dashboard, (2) RID, (3) Virus-CKB, (4) VirHostNet 3.0, (5) HIV Drug Inter- actions, (6) HEP Drug Interactions, (7) HIV and COVID-19 Registry in Europe, (8) United States Swine Pathogen Database, (9) Global.health, and (10) WestNile.ca.gov.

The database comparison included the database name, a hyperlink to the database, and an assessment of whether a login was required. Additionally, the functionality was described, considering aspects such as user friendliness, the availability of a workbench, the ability to work with personal data, tool availability, subjective complexity, the type of downloads offered, and the possibility of accessing all the data. Last, a short description of each database was provided along with the corresponding references.

In the content section, we listed an estimate of the number of families, genera and species covered in the database. Note that only the species estimate (# spec) is included in Table 1. If the database reported the number of virus species or an equivalent numbers—such as number of vOTUs in the case of IMG/VRs—we used that number. If there was no species number or equivalent reported (e.g., for the PaVe), we took the total count of unique ICTV or NCBI Taxonomy virus species names plus “Unclassified <genera>” as the number of virus species. Further details can be found in the individual database paragraphs starting in Section 2.

Regarding the current manuscript, we used ChatGPT (May 24 Version) in compliance with the journal guidelines to help generate most of the text. The writing process can be simplified as follows: 1. We defined a working outline of the contents of the paper. 2. We used each point in the outline as a prompt to generate a paragraph. 3. The paragraphs were edited for flow and scrutinized to make sure the content was as intended. This process was applied to the entire manuscript text, excluding the first paragraph of the introduction, the FAIR criteria paragraphs, the conclusion and the figure and table captions.

**Table 1 viruses-15-01834-t001:** Overview and comparison of recently updated (2022 or later) virus-related databases. A comprehensive description of sequence counts especially in the metagenomic context (vOTUs) can be found in Section 2.3. To obtain more detailed information regarding the download process, please see Appendix A. 
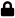
: To access the complete range of features and download data, user authentication is required. #seq-n—number of nucleotide sequences (genes, genomes and/or transcripts) available for bulk download; #seq-p—number of protein sequences (translated genes or open reading frames) available for bulk download; #spec—number of species; –—not known or no access; use—subjective impression of usability (

: Highly user-friendly 

: Easy to use 

: Moderate usability); workb.—workbench (

: yes, 

: no); own d.—own data, the possibility to work with personal data (

: yes, 
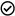
: just Blast, 

: no); compl.—complexity of database: low (
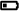
), medium (
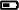
) and high (
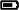
); tools—availability of tools and a quantitative ranking: 
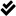
—a lot of tools; 

—available; 
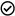
—only few tools; 

—no tools available; F—findability; A—accessibility; I—interoperability; R—reusability (see Appendix A and Appendix A for descriptions (compliance: 

 high 

 medium 

 low 

 no); Down—downloadable via Web (W), FTP (F), and API (A); Click—by one click no data (no), one dataset (one), selected data (sel), or all data can be downloaded (all); Listed in ((

: yes, 

: no): re3da—re3data; Fshare—FAIRsharing.org; DBcom—Database Commons; elixir—ELIXIR bio.tools; NAR—NAR Database list; for VVR: just one of the seven VVR resources is listed.

Name	#seq-n	#seq-p	#spec	Functionality	F	A	I	R	Down	Click	Listed in	Cite
				**Use**	**Workb.**	**Own d.**	**Tools**	**Compl**							**re3da**	**Fshare**	**DBcom**	**Elixir**	**NAR**	
knowledge databases
ICTV	0	0	11,273					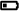					W	all						[29,30]
ViralZone	0	0	7					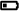					–	–						[31]
VIPERdb	0	0	–					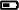					W	one						[32,33]
Virus-Host DB	0	0	16,488					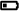					F	all						[34]
genomic sequences databases
BV-BRC	295,306,161	480,376,932	24,824				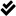	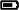					WFA	sel						[35]
NCBI Virus	11,345,662	52,734,161	52,414			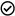	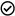	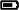					WA	all						[36]
NCBI Viral Genomes	261,488	0	52,414			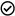	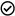	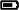	–	–	–	–	WFA	one						[37]
RVDB	8,454,473	0	1,348,976			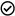	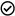	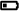			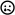		W	all						[38]
VOGDB	11,423	610,152	11,133					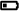					WF	all						–
Virxicon	599,538	0	∼11,273					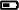					WA	sel						[39]
ZOVER	64,289	64,289	∼1617			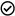	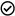	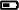					W	sel						[40,41,42]
other—omics databases
IMG/VR 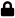	15,677,623	0	9,169,185				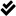	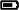					WA	all						[43,44]
MVIP	0	0	77				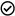	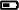					W	one						[45]
Viral Host Range DB	0	0	∼781					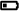					–	–						[46]
				**Use**	**Workb.**	**Own d.**	**Tools**	**Compl**							**re3da**	**Fshare**	**DBcom**	**Elixir**	**NAR**	
virus—specific databases
GISAID 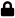	15,320,801	0	3					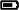					W	sel						[47,48,49]
COVID-19 Data Portal	17,317,438	5340	52					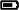					WFA	all						[50]
COVDB	0	0	1				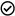	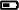					–	–						[51]
LANL HIV Database	1,119,274	74,726	4				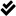	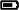					W	sel						[52,53]
EuResist 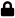	269,205	0	3		–	–	–	–	–	–	–	–	–	–						[54]
HIV Drug Resistance DB	0	0	2				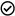	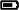					W	one						[55]
HBVdb	132,609	116,318	1					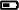					W	one						[56]
PaVE	7228	3985	133				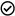	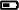					WA	sel						[57]
NCBI VVR	932,938	1,254,767	13				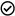	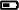					WA	sel			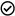			[36]
PSD	4876	0	1					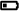				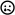	W	all						[58]

Further, we specified the number of nucleotide (#seq-n ) and protein sequences (#seq-p) that were available for browse and bulk download. In more detail, the number of nucleotide sequences referred to the total number of partial and complete nucleotide sequences, which were available as one or more FASTA or FASTQ files. Depending on the database, this number included the genome, gene and/or transcript sequences of viruses. Note that for some databases, plasmids were included in this estimate. The number of protein sequences refers to the number of protein sequences available as one or more FASTA or FASTQ files; this included coding regions, predicted genes (aka open reading frames) and mRNA sequences. Further explanations can be found in the individual database paragraphs starting in Section 2.

In Appendix A, an additional table can be found that further investigates the following aspects: the presence of keyword search functionality, the availability of phrase suggestion, the presence of cross-links to other databases, the ability to share the URL, whether the webpage is generated via an API, the quality of the result table as output, and the availability of the source code. To facilitate a comprehensive analysis of the database catalogs, we included a comparison to determine whether each respective database is included or not. For a detailed examination of the catalogs, see Section 2.4.

### 2.1. Knowledge Databases

These databases play a crucial role in research, facilitating knowledge transfer and providing foundational information. It is important to note that these sources do not directly contain sequences but provide external links to the sequences, instead serving as repositories of centralized knowledge on viruses.



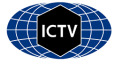



One of the fundamental tasks in virology is establishing a robust taxonomy to facilitate the effective comparison and study of viruses. The International Committee on Taxonomy of Viruses (ICTV) is the global organization tasked by the International Union of Microbiological Societies (IUMS) with developing, refining and maintaining the virus taxonomy down to the level of species [29,30,59]. As of May 2023, the virus taxonomy curated by the ICTV (version: MSL38) comprised 264 families, 2818 genera, and 11,273 species. The taxonomy is periodically updated, with revisions released at least once and up to twice a year. New entries are proposed by the scientific community and reviewed by expert subcommittees within the ICTV. The categorization of virus groups is based on various characteristics, such as genetic material, genome organization, replication strategy, and host range. Notably, the ICTV recently embraced the inclusion of virus groups based solely on sequence information, departing from the traditional reliance on virus morphology. Users can download the entire taxonomy as an Excel sheet or browse through it, available at the visual browser. Links to the genome sequences of exemplars (representatives) and additional representatives of each virus species are provided in the Virus Metadata Resource sheet. In our opinion, the website may profit from a clearer structure and direct download of exemplar and additional virus genome sequences. Helpful are the provided “How-to Videos” as a valuable resource to assist users in effectively navigating the taxonomy.



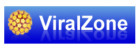



The ViralZone is a powerful and up-to-date online encyclopedic database that provides summarized expert knowledge on various aspects of viruses, including genomic structure, virus replication cycle, host range, virus taxonomy, and molecular biology [31]. Widely embraced by the research community, it has become a trusted resource for obtaining information about specific viruses, serving as a key starting point for addressing novel research questions. In total, it houses detailed descriptions of over 128 families, 567 genera, and 7 virus species (e.g., Influenza A virus and SARS coronavirus 2). Every entry within the database comprises a fact sheet that presents visual representations of the virion and genome, alongside comprehensive details concerning gene expression and replication mechanisms. While the ViralZone itself does not contain nucleotide sequences in bulk download form, it does provide links to protein and nucleotide sequences of reference sequences within the fact sheet. It is a structured, user-friendly and well-connected website, where users and especially virologists can quickly find the information for which they are looking.



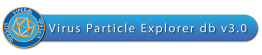



The VIPERdb, a specialized database dedicated to icosahedral virus capsid structures, offers a wealth of information derived from both structural and computational analyses [32,33]. The database provides diverse visualizations on various levels, multiple sequence alignments, relevant publications, and useful tools, such as anomaly analysis and contact finder. Where available, each protein structure page contains a link to the respective protein sequences in the Protein Data Bank, and for those with a related publication, the PubMed id is linked. We did not include these protein sequences in Table 1 because they are not available in bulk download form. The search functionality of VIPERdb allows users to explore virus protein structures based on taxonomic classifications or specific criteria. One limitation of VIPERdb is its focus on icosahedral virus capsid structures. However, there are ongoing efforts to expand its scope, and some helical structures are already included. This comprehensive database encompasses information on 1323 protein structures from 165 viral genera and 75 virus families. With its latest release, VIPERdb introduced a new standalone database on its website, namely the Virus World. It contains a browsable taxonomy and interlinked data (genome type and coat protein sequences) on 211,450 viruses from 1038 viral genera and 158 viral families. Virus World also provides the capsid protein sequences for these viruses in some instances. In our opinion, the search function of VIPERdb could be improved, as users must have prior knowledge of their virus of interest before using it. Further, bulk download and browse options would increase user friendliness. Please note that there was an additional database known as VIPR, which was recently incorporated into the BV-BRC (Bacterial and Viral Bioinformatics Resource Center) as described below.



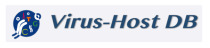



The Virus-Host DB is a comprehensive and manually curated database that links viruses and hosts using pairs of NCBI taxonomy Ids [34]. It includes viruses with complete genomes stored in NCBI/RefSeq and GenBank, with the accession numbers listed in EBI Genomes. Host information is collected from various sources, including RefSeq, GenBank, UniProt, and ViralZone, and supplemented with additional data obtained from literature surveys. The database offers comprehensive information on 15,179 virus species, encompassing scientific names, lineages, Baltimore groups, RefSeq genome sequences, database links, and details of 3791 host species, enabling users to investigate interactions from both virus and host viewpoints. The database is well-interconnected, making it user friendly and valuable for obtaining an overview of interactions. The database contains a limited amount of information, as it focuses on providing specific linkages rather than comprehensive data.

### 2.2. Databases Containing Virus Sequences

Genomic, transcriptomic, and proteomic virus sequences serve as a foundational element for a wide range of virus bioinformatics analyses. For example, phylogenetic analysis typically starts with multiple sequence alignment of a collection of sequences. For host sequences, we refer to additional -omics databases; see below. Sequence databases serve as a critical starting point for examining genetic variations and functional components. When working with all or a significant portion of sequences from a database for further analyses, e.g., in virus ecology, it is important to be aware of the imbalance in virus representation. In other words, the composition of viruses sequences within a database does not reflect the natural occurrence of viruses.



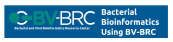



The Bacterial and Viral Bioinformatics Resource Center (BV-BRC) is a recently merged platform that integrates various NIAID-funded pathogen-related resources, including the Virus Pathogen Resource (ViPR), the Influenza Research Database (IRD) and PATRIC (the Bacterial Bioinformatics Database and Analysis Resource). With diverse computational tools, the BV-BRC empowers researchers to analyze and interpret large genetic datasets originating from NCBI GenBank and Refseq (see below) as well as specific projects. Users have the ability to search, browse, download, and analyze a multitude of data types, including metadata, taxonomy, genomes, features (ORFs), proteins, protein structures, domains and motifs, epitopes, and experimental data. It also offers a private workbench for the secure analysis and storage of private datasets. The BV-BRC encompasses integrated datasets from mainly pathogenic bacteria, archaea, viruses, and eukaryotes, allowing users to search, browse, download, and analyze various data types, such as metadata, taxonomy, genomes, features, proteins, and experimental data. Tools and services are categorized into genomics, phylogenomics, protein analysis, metagenomics, transcriptomics, and utilities. The BV-BRC provides access to 295,306,161 virus gene and genome, partial and complete sequences including 9,763,946 genomes/segments, representing 106 virus families, 1946 genera, and 24,824 species. Additionally, there are 480,376,932 virus protein sequences. The BV-BRC categorizes plasmids under the virus category, resulting in the inclusion of plasmid sequences within the overall sequence count. Note that the number of species is higher than that of the official ICTV number because the BV-BRC includes unclassified taxa. Despite the complexity of the platform, efforts have been made to maintain user friendliness and visual accessibility. Workshops and training opportunities are provided regularly to enhance user proficiency in utilizing the database effectively.



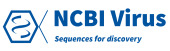



The recently established NCBI Virus interface is a consolidation of various NCBI resources: NCBI Viral Genomes (a former version of NCBI Virus), NCBI Nucleotide (selected for taxonomic classification to viruses), including Refseq, Genbank, Virus Variation Resource, and the old resource NCBI Retroviruses [36]. Virus genome sequences are submitted by users into the public sequence repositories, which are part of the International Nucleotide Sequence Database Collaboration (INSDC). The INSDC collaboration is composed of three organizations: the National Center for Biotechnology Information (NCBI) GenBank, the EMBL-EBI European Nucleotide Archive (ENA) and the DNA DataBank of Japan (DDBJ). These three repositories contain the same data, ensuring data consistency across platforms. The sequences from these repositories are frequently utilized as a starting point by other databases, which then apply diverse analyses or visualizations to further explore the data.

NCBI Virus is regularly updated, with the core component being the GenBank and Refseq sequences and well-curated metadata, and additional features being new analysis or visualization functionalities. As of June 2023, 11,345,662 virus nucleotide and 52,734,161 virus protein sequences were accessible through NCBI Virus. These sequences are linked to the NCBI Nucleotide database, providing extensive metadata, such as organism, host, taxonomy, publication, and organization (e.g., ORF or domains), as well as the corresponding nucleotide or protein sequences. The number of species in the NCBI Virus database, which is 52,414, surpasses the official ICTV count due to the inclusion of unclassified taxa. However, it is important to note that within the broader NCBI GenBank database, there are instances of sequence errors that can potentially contribute to false-positive results in analyses (see Section 3 for more details). NCBI offers the Reference Sequence (RefSeq) collection as a comprehensive, integrated, and well-annotated dataset containing extensive curation and additional data types, including 19,975 nucleotide sequences and 710,847 protein sequences of viruses. The RefSeq are considered high-quality sequences and are extensively utilized by the scientific community.

The search interface of NCBI Virus is user-friendly, and the results are filterable by a wide range of curated metadata, such as taxonomy, length, completeness, host, submitter, genome molecule type, and date. Users can perform a sequence blast or keyword search, with example searches, such as “all viruses” or “bacteriophages”. Based on our experience, datasets containing up to 100,000 sequences can be readily downloaded, offering users a range of options to select from. Additionally, users can conveniently create their own custom FASTA headers. Several tools are available to perform alignments or phylogenetic analyses with selected sequences. Over the years, NCBI Virus resources have evolved, offering enhanced functionality. Compared to the older NCBI Viral Genomes database [37], NCBI Virus is more organized, functional, and visually appealing.The Virus Variation Resource (VVR) covers seven virus species (influenza virus, dengue virus, Zika virus, rotavirus, West Nile virus, MERS coronavirus, and Ebolavirus). Only the sub-database for influenza virus provides extra functionality, such as an annotation tool. NCBI SARS-CoV-2 Resources is another specific virus database for COVID-19 only. In summary, NCBI Virus serves as the go-to resource when working with NCBI virus-related data, offering a visually appealing and user-friendly interface for virus sequence data and metadata.



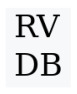



The Reference Viral Database (RVDB) comprises a comprehensive collection of nucleotide sequences, encompassing viral, virus-related, and virus-like sequences (excluding bacterial viruses) [38]. The RVDB provides a curated and a non-redundant subset of GenBank genome sequences, facilitating downstream bioinformatics analyses. The database provides two versions: an unclustered and a clustered version based on sequence similarity. We took the number of sequences available in the unclustered version as an estimate of the number of genomes (8,454,473), while we took the number of sequences available in the clustered version as an estimate of the number of species (1,348,976). Further, old versions of the database are available. Researchers can conveniently download all RVDB sequences in one file, although it should be noted that the large file size (approximately 20 GB) may result in longer download times.

The database is designed with simplicity in mind, offering user-friendly functionality. Additionally, the database provides a BLAST tool for performing sequence searches against different versions of the RVDB, further enhancing its usability.



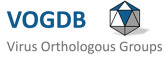



The Virus Orthologous Groups Database (VOGDB) is an automatically updated database that provides virus protein family sequences derived from RefSeq virus genomes. This provides a comprehensive representation of viral protein families as virus orthologous groups (VOG) for comparative virus (meta-)genomics. The VOGDB release vog218 contains 38,109 VOGs derived from 610,152 protein sequences (self-reported at https://vogdb.org/reports/release_stats (accessed on 7 August 2023)) from 11,133 species (estimate extrapolated from the number of species in the corresponding NCBI Refseq release 218), allowing for multiple assignments of the same sequence to different VOGs. While VOGDB currently supports searching for VOGs and provides taxonomic information, the direct downloading of a single VOG is not available. Instead, users can access fileshare platforms, or choose from 11 different (compressed) file formats for their downloads, which may be slightly disorganized. Surprisingly, there have been no publications published by the creators of VOGDB to date.



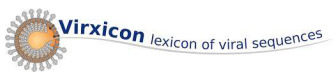



The Virxicon is a centralized knowledge base, gathering information about viruses and their associated sequences [39]. Virxicon is a database that maintains the ICTV virus taxonomy, incorporating virus genome sequences from the NCBI Viral Genomes database and GenBank, and annotating them based on the Baltimore classification system. The database comprises a total of 599,538 genome sequences (the website statistics were not retrievable for the numbers of families, genera or species represented). In their research paper, the authors compare Virxicon with other databases, such as ViralZone, NCBI Virus, and ViPR (now BV-BRC), aiming to combine the strengths of these databases. Virxicon facilitates the bulk download of virus genome sequences with searchable, well-curated metadata, namely Baltimore class, molecular types, and topological resources. However, it is our impression that the database does not provide unique functionality compared to other virus databases. NCBI Virus and BV-BRC provide a larger number of sequences and more extensive functionality related to sequences, including search and tools, while ViralZone serves as a more comprehensive lexicon, including virus sequence download and curated simple metadata. The Virxicon website offers an intuitive and user-friendly interface, providing search and easy access to information.



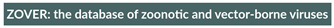



The ZOVER, a comprehensive database of zoonotic and vector-borne viruses, aims to integrate virological, ecological, and epidemiological information to enhance understanding of animal-associated viruses and their significant impact on human and animal health [40,41,42]. ZOVER is a valuable resource, offering a curated subset of NCBI GenBank data, manually collected from the published literature, focused on four specific hosts: bats, rodents, mosquitoes, and ticks. ZOVER was merged from the Database of Bat-associated Viruses and Database of Rodent-associated Viruses. The ZOVER database includes 64,289 gene and genome sequences from 135 genera, combining both protein and nucleotide sequences, making it challenging to differentiate them individually. ZOVER presents data in a well-organized, visualized and user-friendly manner, providing a comprehensive and visually appealing platform for accessing information. ZOVER offers a valuable tool for researchers in the field, as it provides curated and easily accessible data, enhances data visualization, and offers a user-friendly interface for efficient exploration and analysis. Users can easily navigate the database using taxonomy-based searches or various search options, including sequence based, text based or region based.

### 2.3. Omics Databases

The emergence of databases dedicated to -omics data and analyses represents a remarkable advancement in the field of virology. These specialized resources go beyond traditional databases, providing a next-level platform for researchers to delve into the vast realm of -omics datasets and unlock hidden viral treasures. By focusing on -omics data, which encompass various -‘omics’ disciplines, such as genomics, metagenomics, transcriptomics, and proteomics, these databases offer a comprehensive view of the viral world at a molecular level. These databases serve as central hubs for storing, organizing, and analyzing -omics datasets, enabling researchers to explore uncharted territories and uncover previously unknown virus species.



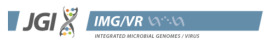



The online platform The Integrated Microbial Genomes/Virus (IMG/VR) provides its own geNomad analysis workflow, with which IMG/VR systematically identifies viral sequences from user-contributed and publicly available datasets, providing researchers with a comprehensive collection of uncultivated virus genomes [43,44] with different levels of confidence for download and analysis. The resource incorporates data from the metagenomic and metatranscriptomic JGI database IMG/M, RefSeq database, and three specific virus databases, while enhancing the annotation process with genome quality estimation, up-to-date taxonomic classification, and microbial host taxonomy prediction. IMG/VR offers users a comprehensive platform with abundant information, analysis tools, and links to sub-databases, including detailed meta-information and statistics for each virus. However, for us, its navigation and accessibility pose challenges, necessitating substantial time investment for users to become acquainted with its features. The database contains an impressive collection of 15,677,623 vGenomes (equivalent to number of nucleotides), organized into 9,169,185 viral operational taxonomic units (vOTUs, equivalent to number of species) across various viral families and genera. Despite its importance in the study of uncultivated viruses (UViGs), users should be mindful of the complexity of IMG/VR and its lack of user friendliness, requiring a login for full functionality and demanding considerable effort to effectively explore and utilize all available features.



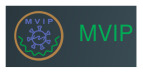



The Multi-omics Portal of Virus Infection (MVIP) collects and analyzes virus infection-related high-throughput sequencing data, integrating comprehensive meta-information [45]. It enables -omics data analysis and visualization, presenting a summary table of samples for specific tissues and viruses. Users can access detailed datasets, including differential expression, pathway enrichment, and alternative splicing, which are downloadable. MVIP provides external resource links and allows user submissions for broader analyses and database enhancement. MVIP offers valuable information and analysis for specific biosamples, driving advancements in the understanding of virus infection, and provides users with the opportunity to suggest biosamples for integration. Currently, MVIP boasts a dataset comprising approximately 6586 sequencing samples derived from 77 distinct viruses, such as SARS-CoV-2, SARS-CoV, dengue virus (DENV), Zika virus (ZIKV), and Influenza A Virus (IAV), across 33 host species, including *Homo sapiens* and *Mus musculus*. MVIP is a visually appealing database that provides comprehensive -omics data analysis capabilities, serving as both a resource for analyzing existing data and a knowledge base for researchers conducting their own sequencing projects, offering insights into the availability of suitable datasets for specific research questions, despite occasional short loading delays.



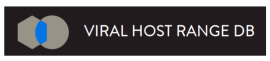



The Viral Host Range Database (VHRdb) is a unique resource that consolidates experimental data on the range of hosts that a virus can infect [46]. Despite the wealth of host-range experiments conducted in laboratories, these valuable data are often inaccessible and underutilized. The VHRdb is an online platform that centralizes experimental data on viral host ranges, allowing users to browse, upload, analyze, and visualize the results. Currently, it contains 17,170 interactions between 776 viruses and 2041 hosts from 20 datasets. Among the 776 viruses in the VHRdb, 303 are linked to the NCBI, representing 279 species from 25 families. The comprehensive overview of virus–host interactions is presented in a visually appealing table, categorizing the relationships into “No infection”, “Intermediate”, and “Infection”. The VHRdb provides extensive and helpful documentation, including quick-start guides, to assist users in navigating and utilizing the database effectively. However, a limitation of the VHRdb is its relatively limited representation of viruses, which are not evenly distributed across various viral families, relying heavily on available studies that may not provide comprehensive coverage. To mitigate this, users can upload their own data for public access or private use. Despite these limitations, the existing studies are visually appealing, allowing for straightforward interpretation and analysis.

#### 2.3.1. Specific Databases

In addition to comprehensive databases, there are numerous virus-specific databases available. If one is working on a specific virus, it is worthwhile to explore specialized databases dedicated to that particular species or group. Here is a brief list of potential databases that primarily focus on a single virus species. One exception is the NCBI VVR database, which specifically addresses seven different viruses as mentioned earlier. For coronaviruses, we rely on the following databases: (1) GISAID, (2) COVID-19 Data Portal, and (3) Stanford Coronavirus Antiviral & Resistance Database (COVDB) [47,48,49,50,51]. For HIV, we have the following databases listed in Table 1: (1) LANL HIV Database, (2) EuResist, (3) HIV Drug Resistance DB, and (4) PSD [52,53,54,55,58]. Due to our inability to access EuResit by the time of submission, the investigation could not be conducted as extensively as with other databases.

For Hepatitis B virus, the dedicated database is HBVdb [56]. And for papillomaviruses, we have the PapillomaVirus Episteme (PaVE) database [57].

#### 2.3.2. Non-Viral Specific Databases

In addition to virus-specific databases, there exist numerous databases that are also important for virus research in the broader fields of biology and genomics. We would suggest also for interested users to keep an eye on initiatives, such as the Global Core Biodata Resources, which seek to identify invaluable, long-term resources for the life sciences.

UniProt [60] provides a vast collection of protein sequences and functional insights, including those from viral sources, enabling researchers to unravel the molecular mechanisms and biological functions of viruses. The Rfam database [61], widely recognized and utilized, encompasses RNA families with detailed sequence alignments, secondary structures, and covariance models, while the Pfam, now InterPro database serves as an extensively employed resource, offering multiple sequence alignments and hidden Markov models for protein families [62]. The NCBI houses additional non-virus-specific databases, such as the Gene Expression Omnibus, which serves as an international public repository for high-throughput functional genomic datasets, or Sequence Read Archive(SRA), a valuable resource that provides access to biological sequence data, fostering reproducibility and enabling new discoveries through dataset comparisons within the research community [63]. Of note is the new NCBI datasets browser (currently in beta version), which provides easy searchable access to different NCBI databases and NCBI Taxonomy via fact sheets. Kyoto Encyclopedia of Genes and Genomes (KEGG) is a comprehensive biological database that represents molecular networks and pathways, facilitates analysis of genomic data, and integrates drug labels and disease databases, making it one of the most widely used resources in the field [64,65,66]. The miRBase is the central repository for microRNA (miRNA) [67]. It enables users to search and browse entries representing hairpin and mature miRNA sequences. Entries can be retrieved by various criteria, and both sequence and annotation data are available for download. The database currently includes 320 precursors and 510 mature miRNAs related to viruses.

#### 2.3.3. Other Databases

Additionally, there exist virus-related online platforms that link together pre-existing tools, databases, and datasets. These websites serve as valuable resources for researchers and practitioners seeking to leverage existing resources and foster collaboration within the scientific community. By linking together disparate resources, these platforms contribute to the dissemination and accessibility of scientific information, promoting the efficient utilization of available resources for further research and innovation. One example is the European Virus Bioinformatics Center (EVBC) website, on which a total of 275 entries are linked, sorted by software type (such as database, command-line tool, or similar), virus family, or functionality [68,69]. Another example is iVirus.us, which provides a platform to access 27 tools and 21 datasets [70,71].

#### 2.3.4. FAIR Evaluation

Many virus databases aim to support the (re)use of virus data and enable processing using machine-learning methods. Both goals can be facilitated by adopting the FAIR principles. We therefore included an evaluation of FAIR properties in our database overview, using FAIR principles checklist. Where a virus database had a table featuring one virus per row, the entries were evaluated as research objects (please refer to the data sources of the FAIR evaluation of the databases in Appendix A). Where available, a virus sequence was considered to be “data”. Note that some databases in the list were therefore excluded from the FAIR evaluation because of a lack of a comparable research object. The databases that did not have comparable research objects were NCBI Viral Genomes (due to being a central website linking to different resources) and the EuResist database (to which we did not have access by the time of submission). The FAIR scores are based on presence/absence for each of the checklist criteria as was done previously in the context of data deposition of nuclear magnetic resonance data [72]. The scores are out of four for the subcriteria in findability, accessibility and reusability, and are out of three for those of interoperability. A more complete description of the FAIR Principles checklist can be found in Appendix A.

In general, the FAIR scores of the content of the active databases reviewed here (summarized in Table 1 and the full table available in Appendix A) ranged from less FAIR for the smaller or older databases and more FAIR for the larger and newer databases. An important component of the findability score is the assignment of a database-given global and persistent identifier; while the large platforms such as BV-BRC and IMG/VR featured this, the smaller databases such as HBVdb often used an external id, e.g., the NCBI Accession ID or TaxIDs. This might be due to the differing aims of the virus databases, as some are focused on data reuse and machine-readability, while others may have simpler goals, such as cross linking available knowledge. Accessibility for the databases was generally positive owing to web-accessible links and straightforward download options (see also Appendix A). Further, the overall low score for interoperability reflects the lack of standards for all virus metadata; while there exist clear ontologies, e.g., for clinical data (as for the HIV drug resistance DB) or for pathogenic virus metadata (see the Genomic Standards Consortium (GSC)), this is not yet the case for metadata for all viruses. This is currently a target for various groups, such as the GSC (which is responsible for the minimum information about sequencing standards that are used by the INSDC repositories), the Gene Ontology consortium, the Genomes Online Database (GOLD) which complements the IMG databases of the Joint Genome Institute JGI, and other efforts such as those of Bernasconi et al. with the viral conceptual model [73,74]. This shows that community-wide metadata standards are poised to improve interoperability in the near future. Last, the reusability of many of the virus databases would benefit from the inclusion of formal licenses describing the reuse of their data (see Choose A License). Overall, this FAIR evaluation was a first for virus databases and highlighted several areas for improvement.

### 2.4. Catalogs of Databases

To assist users in selecting appropriate databases, scholarly journals and other entities have established catalogs that employ various criteria for indexing databases based on different criteria to improve their findability and accessibility. Here, we describe five catalogs of databases: (1) re3data.org, (2) FAIRsharing [23], (3) The Database Commons [24], (4) ELIXIR bio.tools [25], and (5) NAR database list [26] (see Figure 1). We analyzed a range of entries, narrowing down to virus-specific databases, categorizing them based on their up-to-date status and relevance to COVID-19, while excluding non-virus databases that did not meet the criteria.



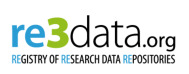



The re3data.org website is a web-based registry that facilitates data discovery, access, and sharing for researchers. Its comprehensive metadata on data repositories allow researchers to identify repositories that align with their specific data management needs. The platform has a particular focus on the FAIR principles. There are 3125 entries on this platform, of which 2181 are databases or scientific and statistical data formats in terms of content types. Among them, there are 186 virus-related entries identified using the search term “virology”, of which only 24 are virus-specific, and 17 are considered up to date. Nine of these databases are extensively described in our curated Table 1, seven are dedicated to coronavirus research (see Appendix A, and one database, namely WestNile.ca.gov, is excluded due to its narrow focus.



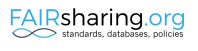



The online platform FAIRsharing is designed to enhance the visibility of scientific data standards, databases, and policies for the scientific community. The platform includes a registry of data standards, databases, policies, collections, and organizations that detail each resource, such as its scope, history, and adoption status. In total, 3888 entries are listed in the registry, of which 2032 are repositories or knowledge bases. Among these, 112 are virus-related (identified using the keyword “virology”). However, only 62 of these resources are virus-specific, and only 41 are up-to-date. These 41 resources can be further classified into the 13 that are listed in our curated Table 1, the 24 related to coronavirus data (see Appendix A), and the 4 we excluded: (1) HIV Drug Interactions, (2) HEP Drug Interactions, (3) Global.health, and (4) HIV and COVID-19 Registry in Europe. These databases were excluded due to their restricted focus, such as focusing only on drug interactions of a particular virus or containing primarily epidemiological data, which did not align with our definition of a comprehensive virus database. Additionally, one of the databases resembled more of a network than a traditional database.



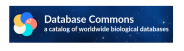



The Database Commons is a curated catalog of biological databases that organize databases based on data type, species, and subject matter. It provides detailed metadata for each database, including name, URL, description, hosting institution, and contact information. Within the Database Commons, there are currently 5902 entries listed. Among them, 355 databases fall under the “Data Object” virus category. Of these, 146 are virus-specific, and 36 are considered up to date. These 36 databases can be further categorized as the 17 listed in our curated Table 1, 16 coronavirus databases (see Appendix A), and 3 other databases (Disease Monitoring Dashboard, RID, and Virus-CKB). The additional databases were excluded due to their specific nature, such as being more tool-oriented or containing limited data with only two tables rather than meeting the criteria of a comprehensive virus database.



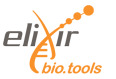



A comprehensive registry of bioinformatics resources is established through a community-driven curation effort supported by ELIXIR, a (ELIXIR). ELIXIR bio.tools serves as the dedicated registry within this infrastructure, ensuring the sustainable upkeep of the curated information [25]. Collaborative curation, tailored to local needs and facilitated by a network of partners, enables the continuous development and accessibility of this valuable resource. In total, there are over 28,211 resources listed in the registry, including various tools. Among them are 3664 databases, and a search using the keyword “Virology” identifies 44 databases in this category. Out of these, 42 databases are virus-specific, with 11 being up-to-date. Eight of these virus-specific databases are included in our curated Table 1. Additionally, we identified two up-to-date coronavirus-specific databases and one particular database, namely the United States Swine Pathogen Database, which we excluded.



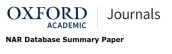



To our knowledge, the Nucleic Acids Research Journal Database Summary Issue NAR is the oldest known list of databases. Published annually, it provides descriptions of new and updated databases that contain nucleic acid and protein sequences and structures [26]. The NAR provides the links for these databases at the Molecular Biology Database Collection. These databases are categorized into genomics, transcriptomics, proteomics, metabolomics, and structural biology. Presently, it includes a total of 1965 databases. Each database is described in detail, including its scope, content, features, relevant citations, and links to access the resource. The most recent issue from January 2023 lists 32 databases in the “virus genome database” category. Among them, 9 are considered up to date and are included in Table 1.

In conclusion, despite the availability of database catalogs that assist researchers in finding relevant resources, there are still challenges and limitations to address. These catalogs lack virus-specific content and often do not reflect the current status or usability of the databases. Furthermore, there is a need for better metadata standardization and information on the reliability and quality of the databases. Although these catalogs serve as a starting point, they may not provide comprehensive and detailed information for researchers to make informed decisions about utilizing the databases effectively.

## 3. Evaluation of Errors in the NCBI and BV-BRC

Despite diligent curation, databases like the NCBI Nucleotide database often harbor errors, including those arising from user-generated data. These errors extend beyond user mistakes and stem from the need to adapt databases to rapidly evolving scientific fields like virus taxonomy. Viromics poses challenges, such as the absence of a universal viral gene, the facilitation of horizontal gene transfer, and the need for specific data models and standards. Addressing these challenges requires specialized protocols for RNA and DNA viruses and mitigating experimental biases related to enrichment methods [75]. As user-friendly pipelines for comparative genomics become more prevalent, the quality of viral sequences from databases used as input becomes crucial for reliable and accurate bioinformatic analyses. Incorrect input data can undermine the validity of downstream results, regardless of the pipeline employed. Therefore, it is essential to critically scrutinize and validate the outputs obtained from these pipelines, mainly when errors exist in the utilized databases. Taking NCBI and BV-BRC as an illustrative example, it is important to acknowledge the various types of errors that can occur in databases, namely (1) taxonomy errors, (2) naming and labeling errors, (3) missing information, (4) sequences errors, (5) wrong orientation, and (6) chimeric sequences.

### 3.1. Taxonomy Errors

The NCBI Taxonomy [76] incorporates most of the ICTV entries, supplemented by a number of additional taxa; see Figure 2. ICTV defines only approximately one-fifth of the species mentioned in NCBI. In rare cases, inconsistencies may occur, but they are typically resolved in subsequent updates. The higher number of species in the NCBI Virus database compared to the official ICTV count is primarily attributed to the inclusion of unclassified and unverified taxa. As outlined above, the ICTV serves as the authoritative source for virus taxonomy. Within the ICTV, new taxa are evaluated by an expert committee, and the taxonomic classification extends only up to the rank of species. More detailed subdivisions, such as subspecies or lineages, are not determined by the ICTV but are entered by the submitter to NCBI. TaxIDs are commonly used for taxonomic groups in databases like NCBI and UniProt but not in ICTV. The NCBI Virus database houses around 53,000 virus species, with nearly 1.5% having more than one TaxID assigned (e.g., Lomovskaya virus with 6 TaxIDs); see Figure 2. Multiple TaxIDs in NCBI for certain viruses result from the ongoing refinement of virus classification, including species mergers and new TaxID assignments. An additional challenge arises in cases where assigning two TaxIDs to a specific sequence is needed as seen in studies involving integration sites, where a sequence represents both the virus and the host [77]. Implementing a mechanism to accommodate such dual assignments would better reflect the intricacies of these scenarios.

### 3.2. Naming and Labeling Errors

Within the realm of virus labeling, an ongoing systematic error can be observed, where users label sequences as “complete genomes” despite the partial or complete absence of untranslated region (UTR) sequences [78], for example, the Hepatitis C virus sequence (EU255989.1), even though the UTR sequences are missing. The widespread occurrence of this error is exemplified by the Hepatitis C virus in the BV-BRC, where a search for complete genomes yielded 3676 results. Upon closer examination, 69 specific sequences did not meet the expected length of 8500 bp, indicating that these are incomplete genomes.

The length of the virus genomes alone is not sufficient to demonstrate the extent of the problem. To assess the prevalence, a blastn (version: 2.5.0, E-value $10^{-4}$) search was conducted on the 3676 listed complete Hepatitis C virus sequences using representative 5′UTR sequences, revealing that only approximately 80% of the sequences contain the conserved 5′UTR. Inconsistency in the usage of completeness terminology across databases is another issue. When examining all sequences labeled as complete in the NCBI Virus database, it was discovered that 1.95% of them are classified differently in terms of completeness in the BV-BRC database.

Incorrect metadata can extend beyond the completeness metadata field as exemplified by the case of sequence AJ000888.1. Here, the “source” field is incorrectly identified as *Hepacivirus hominis* despite being of human origin.

Another important field containing errors is the “organism” field, which is used by many secondary databases. Differences in this field mainly result from historical changes or typos. Virus names are chosen with the most current knowledge available with each release of the ICTV taxonomy. Ongoing changes to the virus taxonomy mean that the corresponding names in the NCBI are updated yearly with the ICTV taxonomy. Approximately 15% of virus species described in NCBI Taxonomy have multiple names, with the virus species *Gallid alphaherpesvirus 2* having the most additional names at 16. These include previous official names, common names and different spellings/punctuation. Appendix A lists different types of submission and naming errors. Possible contributions to the presence of multiple names of virus species could be the lack of best practices for naming virus species and typos upon submission, which are not curated. Due to the relatively low volumes of sequences deposited in the past, best practices were not needed; however, this is becoming a priority with increasing amounts of deposited data [79]. With every sequence upload, potential new errors may be added, e.g., by filling the organism field incorrectly.

The naming of fasta headers can also contribute to confusion. Sequences can contain cryptic names (e.g., JB021961.1-Sequence 12 from Patent EP2495325, A12995.1-Fragment Ba131, or E04227.1-cDNA to satellite RNA). This can cause difficulties in forward analysis. A more user-friendly alternative is implemented in NCBI Virus, where the user can build the name of the sequence from predefined properties independently (e.g., accession, species, and length of sequence).

### 3.3. Missing Information

To maximize the utility of genomic sequences for various purposes, it is essential to collect metadata on the properties of the pathogen and make them available in organized, clear, and consistent formats. Several studies have focused on identifying the necessary minimum of metadata [20,80]. In this specific case, the variation among publications and their focus on different types of data make it challenging to reach a clear consensus. Nonetheless, we propose four metadata groups to consider: collection attributes (year, source, country, and host), database crosslinks (GenBank accession, taxon linage ID, and publication), species variations (lineage, strain, and subtype), and sequence information (segment information, genome quality, and genome completeness status).

Almost a quarter (28.57%) of the 9,763,946 genomes/segments described in the BV-BRC are missing more than half of the described metadata above. Astonishingly, for only 68.12% of the sequences, lineage information is provided (see Appendix A for detailed findings). The annotation of features, such as genes, UTRs, and CDSs, also plays a role and can be erroneous or missing. This is exemplified by the case of sequence CS179664.1, a Hepacivirus hominis sequence, where the feature annotations are found to be entirely missing.

To counteract missing or incorrect metadata, several guides and tools are available to help explain how to submit data to the INSDC repositories (see Submitting Data to ENA and NCBI GenBank Submissions, the NCBI Submission Help Page and the NCBI Virus Submission Help Page) [81,82,83]. Relevant information, such as sample collection, sequencing methodology, and bioinformatic procedures, should be noted [84,85,86,87]. Assembly strategies have been shown to significantly affect the resulting sequences, emphasizing the importance of including this information [84,85,86,87]. Guidelines are available to help ensure verification, addressing the issue of missing or incorrect metadata [81].

### 3.4. Sequence Errors

Likewise, nucleotide or protein sequences in NCBI and other databases can contain errors; see Figure 3.

In addition to the challenge of differentiating sequencing errors from the impact of rapid viral evolution, in some cases, sequences lack even any meaningful sequence information. A striking illustration of this is observed in the sequence of 1PFI_C-Chain C, PF1 VIRUS STRUCTURE, which is composed of a solitary N nucleotide.

Furthermore, errors in (reference) sequences can have a cascading effect on subsequent sequences, e.g., during reference-based assemblies, potentially propagating inaccuracies throughout computational downstream analyses. These errors, combined with methodological challenges in genome assemblies, further emphasize the need for scrutiny and validation of genomic data to ensure the reliability of research findings [84,87,88]. Alternately, this serves as an extra measure of caution for users to perform a quality control check when conducting bioinformatics analyses using existing datasets.

### 3.5. Sequence Orientation Error

Viruses have diverse genomes and are classified into different Baltimore classes based on their genome. The NCBI default for user-uploaded virus sequences is the positive sense strand, which is counter intuitive, for example, for viruses in Baltimore class 5 (negative single-stranded viruses).

The uploading of sequences in the wrong orientation as seen in examples like the Orthomarburgvirus marburgense isolate sequence KU059750.1, leads to misinterpretation and further complications. This issue is exacerbated by the fact that many tools do not verify the correct orientation, resulting in nonsensical output that can misguide subsequent analyses. This issue extends to the annotation of functional units, including proteins, which are also provided in reverse order. To identify incorrectly uploaded sequences in terms of orientation, the ORF density was used, which refers to the number of open reading frames (ORFs) found on a particular strand. Out of the 3676 Hepatitis C virus genomes labeled as complete in the BV-BRC, a total of 205 sequences exhibited discrepancies in ORF density, indicating potential errors in their orientation. This highlights a systematic problem in the database.

### 3.6. Chimeric Sequences

Another type of error is the presence of chimeric sequences, where a portion of the sequence is derived from the virus, while another part originates from a non-viral source (e.g., the host). It is important to note that such chimeric sequences can arise due to biological factors or process errors, such as assembly mistakes. In extreme cases, this leads to thousands of hits being found in the host genome with a viral sequence, although the originating sequence part is not viral.

One example highlighting this issue is observed in the last 280 nucleotides of a specific Zika virus sequence KY766069. A comparative alignment between this sequence and the two other RefSeq Zika virus sequences clearly demonstrates that the 3′ end of KY766069 does not originate from the Zika virus as illustrated in Appendix A.

The Zika virus sequence (KY766069) exhibits a fragment of the AluSx repetitive element in its 3′ end, resulting in over 200,000 false positive hits in a blastn search (version: 2.5.0, E-value $<10^{-4}$) [89] on the human genome (hg38.p13). The inclusion of a human partial sequence within the Zika virus sequence is likely a result of a sequencing or assembly error. This highlights the impact that a single chimeric sequence within a dataset can have on subsequent results or outputs.

In summary, it is crucial to acknowledge the presence of errors in databases like the NCBI Nucleotide database to ensure reliable and accurate downstream analyses. Errors can arise from various factors, including taxonomy inconsistencies, naming and labeling errors, missing information, sequence errors, wrong orientation, and chimeric sequences. These errors can undermine the validity of downstream results, and they highlight the need for the critical scrutiny and validation of data. Addressing these errors is essential to maintain the integrity and reliability of virus databases for scientific research.

## 4. Outlook and Conclusions

Here, we provided a comprehensive assessment of active virus databases, defining these as any database last updated in 2022 or later which contains virus-related research data. Our list of 24 databases was compiled through an exhaustive search of active virus databases from two previous reviews of virus databases as well as five catalogs. For the first time, our review includes a thorough evaluation of database usability and contents—including the number of species and sequences contained therein—as well as a FAIRness comparison. We hope this overview will help guide prospective users of these databases. We refrain from suggesting a particular database for use because this highly depends on an individual researcher’s needs. For the knowledge databases that cover a broad spectrum of viruses, we provide a detailed overview and offer suggestions for their potential applications.

The content of the virus databases vary widely depending on the scope, number of data types and tools, and number of virus species. Here, we presented in detail 4 knowledge databases, 7 sequences databases, 3 -omics databases and 10 databases focusing on specific viruses. The databases feature a range of -omics datasets integrated and in combination with sampling, host, collection, environmental and other metadata. In terms of virus species represented, several are focused only on one or a few species (our so-called “specific virus databases”), while on the other end of the range, the large databases feature upwards of 1 million species. The virus databases also vary in terms of usability, with larger databases sometimes presenting a more complex user experience, in which it takes longer to orientate oneself, although some smaller databases could also improve their usability. In general, the databases we listed here are good in terms of findability and accessibility, but we found that community-wide metadata standard development and explicit listing of formal data usage licenses would improve interoperability and reusability. Future database reviews could include an evaluation of the accessibility of metadata standard definitions (in other words, descriptions of database fields) and connection to other databases (e.g., how well links to other databases are incorporated).

We found that researchers might encounter challenges when using existing catalogs of databases for virus-related databases due to limited virus-specific content and searchable metadata. To address this, efforts are ongoing to curate virus-specific database lists (as in the virus subsection of the NAR database list) and to improve domain-specific metadata (which is a goal of re3data, FAIRsharing, the Database Commons and ELIXIR bio.tools).

In part due to FAIRification, we expect that these database catalogs will only improve. We suggest that these catalogs can serve as a useful starting point for any researcher, which can complement curated collections, review articles and word of mouth within specific disciplines specifically for searching a particular database, for example, a particular virus or a certain data type. While finding the right database for one’s research needs may be challenging, using these existing resources is crucial for modern virus research, especially due to the high volume of multi-dimensional data available on viruses. We suggest that the landscape of virus databases be regularly evaluated, and for core virus databases to be included in at least university-level virus-related courses when relevant (e.g., NCBI Virus).

Recent sequencing advancements and the growing interest in viruses within the field of virus bioinformatics have led to rapid changes in the virus database landscape. We illustrated this rapid development by referencing the reviews of Sharma et al. 2015 and Mcleod and Upton 2017 of virus databases and tools [21,22], of which only 22% and 23% of the virus databases listed in either review are still active.

Ensuring longevity for virus sequence databases includes FAIR principle-like criteria in addition to regular maintenance and updates, the creation of backups and archives, collaboration, funding, and trust, and usage by the community [17,18]. For databases, improving FAIRness could help to foster these content-related aspects and contribute to longevity. Moreover, funding plays a large role in the staying power of a database, which in turn is influenced by the database’s ease of use and content quality. Ensuring data and metadata quality is a key concern, and efforts are being made by teams at the INSDC repositories and other institutions to curate existing metadata. We underline the need for clear data submission guidelines and the inclusion of curated sequences together with the regular updating and removal of outdated or redundant sequences. Database content errors can arise from various factors, including taxonomy inconsistencies, naming and labeling errors, missing information, sequence errors, wrong orientation, and chimeric sequences, which can be attributed to either user input or inherent database discrepancies. These errors can undermine the validity of downstream results and highlight the need for the critical scrutiny and validation of data. Addressing these errors is essential to maintain the integrity and reliability of virus databases for scientific research.

In conclusion, the databases listed here represent the current knowledge of viruses, and the current review will help aid future users find databases of interest based on content, functionality, and scope. The use of virus databases is integral to gain new insights into the biology, evolution, and transmission of viruses, and develop new strategies to manage virus outbreaks and preserve global health.

## Figures and Tables

**Figure 1 viruses-15-01834-f001:**
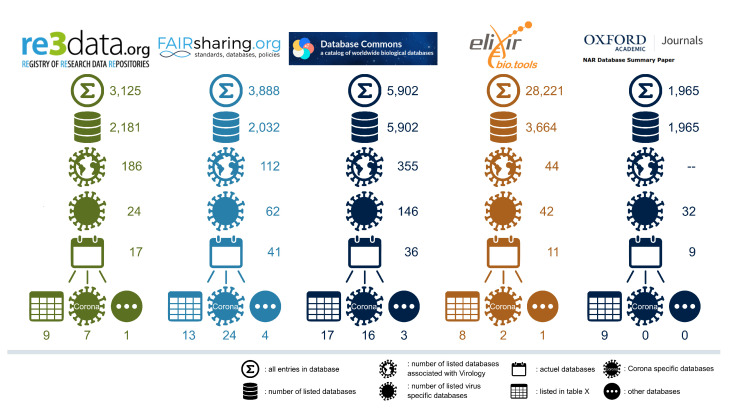
Comparison of five catalogs content: (1) re3data.org, (2) FAIRsharing [23], (3) The Database Commons [24], (4) ELIXIR bio.tools [25], and (5) NAR database list [26]. Among all up-to-date virus-specific databases, they were categorized into three groups: inclusion in our curated Table 1, exclusive focus on coronavirus (see Appendix A), or non-corona-related databases mentioned in the text; see Section 2.4.

**Figure 2 viruses-15-01834-f002:**
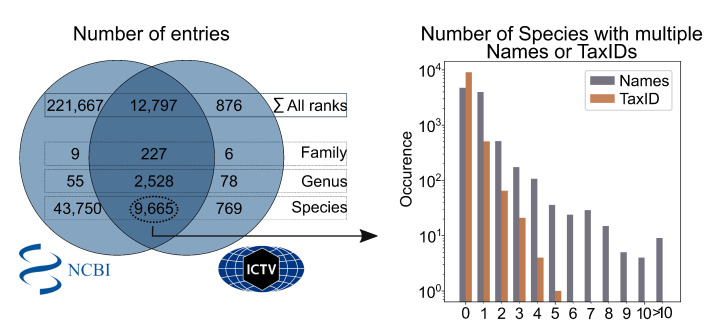
Comparison of the ICTV or NCBI Taxonomy rank names highlighting the existence of multiple names and TaxIDs. (**Left**): The Venn diagram depicts the number of unique names for virus taxonomic ranks in the ICTV or NCBI Taxonomy, or those which occur in both databases for the sum of all taxonomic ranks as well as for the family, genus and species rank names. (**Right**): The virus species names shared between both the ICTV and NCBI Taxonomy are listed on the right panel. For these shared 9665 species names, we examined the occurrence of multiple names and TaxIDs. The *y*-axis is presented in a logarithmic scale. Entries with a value of 0 indicate no presence of additional names or TaxIDs.

**Figure 3 viruses-15-01834-f003:**
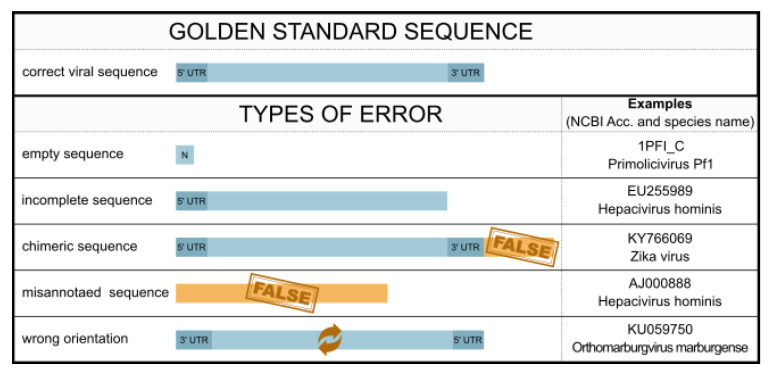
Various sequence-based errors. An artificial, correctly labeled viral sequence serves as an example of a golden standard sequence.

## Data Availability

Data are contained within the article or Appendix A. The data presented in this study are available in the Appendix A.

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
