# Peer review of "Navigating the Landscape: A Comprehensive Review of Current Virus Databases"

_viruses, 2023, doi:10.3390/v15091834_

Round 1
Reviewer 1 Report
The manuscript by M. Ritsch et al. reviews the databases that contain data on viruses. The review thoroughly analyses the contents, tools and other features in these databases. Furthermore, the authors evaluate in detail and discuss the functionalities, compliance with FAIR principles and common errors in a number of virus databases. Importantly, the authors explored the current state of affairs, for instance, indicating update status and availability of analyzed databases. This is a timely overview of available resources in this field, which would be certainly used as a reference material by many researchers. The text is clear and well-written.
Minor remarks:
1. References with numbers starting with 91, referred to in the Supplementary Tables S1 and S2, were omitted from the materials available for reviewing.
2. References 7 and 21 are identical.
Author Response
1. We thank Reviewer 1 for the nice comments. We note that the availability of the references after 91 as in comment 1 is a restriction from the journal, and that the full list is available upon request to the editor.
2. We have removed the duplicate reference 21.
Reviewer 2 Report
First of all, this is a very useful review. The current manuscript is a very good starting point in order to have an overview of virus related databases. Several of the comments on different databases are very valuable and I completely agree with the sentiments. Like “In other words, the composition of viruses sequences within a database does not reflect the natural occurrence of viruses.“, because sometimes the interpretation of data(base) analyses results do not take database bias into account.
Visualisation of the results is very well resolved and understandable.
A few rather technical points:
In several places “ELEXIR bio.tools” should most likely be ELIXIR bio.tools”.
In the section ‘Virus Particle Explorer’ is the sentence: “Each of the 1,332 structures are linked to their respective protein sequences on PubMed.“ Most likely Pubmed is here a typo!?
In Table 1, the exact meaning of a number of species (#spec) have to be specified or the numbers (at least in the ‘virus-specific databases’ section) need to be checked. In PaVE the number of species (species as defined in NCBI taxonomy or ICTV) is definitely higher than 1. I haven’t tested other ‘virus -specific datasets’ in that aspect.
Please check the use of ‘IAV’ and ‘Influenzav Virus’ throughout the manuscript. Is this what the author’s intended?
Please check the references 7 and 21. They seem to be identical.
The probable audience of this manuscript is supposed to be wider than experts on virus bioinformatics. So, in Table 1 it should be explained why #seq-p is lower than #seq-n. It seems that for different databases the reasons are different, but I might be wrong here. The reasons should be explained either directly in the table or referred to in supplementary material.
Hopefully the audience of this article is very wide, so I recommend to specify the use of terms or to explain the terms in the supplementary - nucleotide sequences, genomes, genome sequences. I may be wrong, but in some databases ‘number of sequences’ = ‘number of (near)complete genomes’, but in others they referred number of distinct nucleic acid molecules. Maybe it would be useful if these terms are explained and then used consistently.
In chapter ‘3.1. Taxonomy errors’ the authors should note that ICTV do not manage with taxonomic levels behind ‘species’ and that ICTV taxonomy is based on criteria followed by a committee of expert and (as far as I know) the NCBI taxonomy sometimes depends on submitter.
Please explain the inconsistency in Fig.2. The number of species currently in ICTV and NCBI is 9665 as we see left on Fig. 2 (surrounded by dashed oval). On the Figure legend we see that 8163 species are shown in right panel. Why this difference? Could you please also specify how the ICTV and NCBI taxonomy species are merged? Using species names?
From the point of view of the history of virology, it is a little bit unfair to call the existence of multiple names (from pre-sequencing era) errors.
The following is not a criticism for the current manuscript, but may be after five years when the authors repeat similar examination. It would be reasonable to evaluate the following criteria/aspects as well:
aa) How easy (or at all) is it to find the description of the (database) fields;
bb) How EASY is it to connect by unique identifier to other databases and the correctness of the linking.
Considering these aspects (also covered to some extent in this manuscript) it would make the first steps in virus bioinformatics for virologists (as well for informaticians) easier. This would help to make the viral bioinformatics ecosystem more interconnected and usable.
Round 2
Reviewer 2 Report
Thank you very much for the improvements.
Please correct the use *'STab.??' and 'SFig' with proper name and number through the manuscript. As well as citation 80 in text (currently in form of '[80? ]' .
Author Response
Thank you very much for the improvements.
- We thank the Reviewer in turn for their feedback.
Please correct the use *'STab.??' and 'SFig' with proper name and number through the manuscript. As well as citation 80 in text (currently in form of '[80? ]' .
- We corrected the supplementary material references to the correct form
- We added the references from the supplementary material to the main document and fixed the citation mentioned.